# Chronic Nicotine Consumption and Withdrawal Regulate Melanocortin Receptor, CRF, and CRF Receptor mRNA Levels in the Rat Brain

**DOI:** 10.3390/brainsci14010063

**Published:** 2024-01-09

**Authors:** Oguz Gozen, Buket Aypar, Meliha Ozturk Bintepe, Fulya Tuzcu, Burcu Balkan, Ersin O. Koylu, Lutfiye Kanit, Aysegul Keser

**Affiliations:** 1Department of Physiology, School of Medicine, Ege University, 35100 Izmir, Turkey; oguz.gozen@ege.edu.tr (O.G.);; 2Center for Brain Research, Ege University, 35100 Izmir, Turkey

**Keywords:** nicotine, withdrawal, CRF, CRF1 receptor, MC3R, MC4R, mesocorticolimbic system

## Abstract

Alterations in the various neuropeptide systems in the mesocorticolimbic circuitry have been implicated in negative effects associated with drug withdrawal. The corticotropin-releasing factor (CRF) and α-melanocyte-stimulating hormone are two peptides that may be involved. This study investigated the regulatory effects of chronic nicotine exposure and withdrawal on the mRNA levels of melanocortin receptors (MC3R, MC4R), CRF, and CRF receptors (CRFR1 and CRFR2) expressed in the mesocorticolimbic system. Rats were given drinking water with nicotine or without nicotine (control group) for 12 weeks, after which they continued receiving nicotine (chronic exposure) or were withdrawn from nicotine for 24 or 48 h. The animals were decapitated following behavioral testing for withdrawal signs. Quantitative real-time PCR analysis demonstrated that nicotine exposure (with or without withdrawal) increased levels of CRF and CRFR1 mRNA in the amygdala, CRF mRNA in the medial prefrontal cortex, and CRFR1 mRNA in the septum. Nicotine withdrawal also enhanced MC3R and MC4R mRNA levels in different brain regions, while chronic nicotine exposure was associated with increased MC4R mRNA levels in the nucleus accumbens. These results suggest that chronic nicotine exposure and withdrawal regulate CRF and melanocortin signaling in the mesocorticolimbic system, possibly contributing to negative affective state and nicotine addiction.

## 1. Introduction

Chronic nicotine exposure produces neuroadaptations that affect various neurotransmitter systems within the mesocorticolimbic system [1]. These neuroadaptations underlie the transition to nicotine addiction. When nicotine is removed, drug-induced neuroadaptations cause a withdrawal state, which presents with a group of affective and somatic signs. Withdrawal signs reflect a negative emotional state, which underlies the transition to addiction through the development of negative reinforcement [2,3]. During drug withdrawal, brain stress systems are activated and produce a negative affective state associated with depression-like behavioral responses. In rodents, cessation of nicotine reduces locomotor activity [4], increases immobility time in the forced swim test [5], and elevates the brain reward threshold in the intracranial self-stimulation test [6]. In other studies, nicotine withdrawal was shown to reduce tonic dopamine (DA) neuron activity in the ventral tegmental area (VTA) [7] and extracellular DA levels in the nucleus accumbens (NAc) [8,9]. Decreased DA activity in the mesolimbic system may be involved in the mediation of withdrawal signs [10].

Alpha-melanocyte stimulating hormone (α-MSH) is derived from the precursor polypeptide pro-opiomelanocortin (POMC) and exerts its effects in the brain mainly through melanocortin 3 and 4 receptors (MC3R and MC4R). MC3R has a narrow distribution in the brain, found at highest density in the VTA and hypothalamus [11]. MC4R is more broadly distributed, with highest expressions in the striatum, lateral septal nucleus, and hypothalamus [12]. Intracerebroventricular (ICV) injections of α-MSH produce depressive-like behavior in the Porsolt test and anxiogenic-like behavior in the elevated plus maze (EPM) test [13,14]). Goyal et al. [13] showed that ICV injection of an MC4R antagonist elicited an antidepressant-like effect in the Porsolt test. In parallel, Lim et al. [15] reported that chronic stress-induced increases in anhedonia were prevented by depleting MC4Rs in the NAc. The effects of MC4R signaling on withdrawal-induced behavior were also examined in several studies. ICV administration of MC4R antagonists was shown to block the anxiety-like behavior induced by ethanol withdrawal [14] and prevent reinstatement of nicotine seeking triggered by footshock stress [16]. On the other hand, MC3Rs are indicated in reward-related behavior. Injection of α-MSH into the VTA, which has high MC3R expression, increased DA release in the NAc [17] and enhanced DA-dependent behaviors (locomotor activity and grooming) [18]. MC3R signaling is also indicated in drug-induced reward. Intra-VTA infusion of an MC3R agonist increased the motivation for sucrose [19]. Another study demonstrated increased MC3R binding in the NAc in alcohol-preferring AA rats [20].

Corticotropin releasing factor (CRF) is accepted as a mediator of the brain stress systems. CRF and its type 1 receptor (CRFR1) are widely expressed in the mesocorticolimbic system, whereas the type 2 receptor (CRFR2) has more limited expression [21,22]. ICV administration of CRF induces conditioned place aversion [23] and decreases open arm preference in the EPM [24]. Similarly, infusions of CRF and CRF receptor agonists into the central nucleus of amygdala (CeA), lateral septum, and medial prefrontal cortex (mPFC) increase anxiety-like behavior in the defensive burying and EPM tests [25,26,27] and produce conditioned place aversion [28]. Enhanced CRF signaling in the mesocorticolimbic system is one of the neuroadaptations triggered during chronic exposure to drugs of abuse. Consistent with this, some previous studies showed that exposure to chronic nicotine and induction of withdrawal increased CRF mRNA expression in the VTA and CeA [29,30]. Moreover, Grieder et al. [30] showed that downregulation of CRF mRNA in the VTA blocked the anxiety and conditioned place aversion induced by nicotine withdrawal. CRFR1 antagonists administered systemically or by ICV injection prevent elevation in brain reward thresholds [31] and decreased anxiogenic-like behavior in the defensive burying test [26] and EPM tests [32] during nicotine withdrawal. Evidence suggests that CRFR1 and CRFR2 play opposing roles in the regulation of the negative affect. ICV administration of CRFR2 agonists decreased the anxiety- and depression-like behavior induced by nicotine withdrawal [33].

In the present study, we examined the effects of chronic oral nicotine exposure and withdrawal on the mRNA expressions of melanocortin receptors (MC3R and MC4R), CRF, and CRF receptors (CRFR1 and CRFR2) in the mesocorticolimbic system. To the best of our knowledge, no previous study has examined nicotine regulation of melanocortin receptors in the mesocorticolimbic system. Although several studies have investigated the change in the mRNA levels of CRF and its receptors in the mesocorticolimbic system during chronic nicotine exposure and withdrawal [29,30,34,35], their findings are contradictory. The presence of such regulatory effects would indicate a role for melanocortin and CRF signaling in nicotine reward and withdrawal-induced negative effects.

## 2. Materials and Methods

### 2.1. Animals

The study was performed with adult (4–5 months old) male Wistar rats. A total of 40 rats were purchased from the university animal breeding facility. The rats were pair-housed in standard plastic cages with food and water ad libitum, with a 12:12 h light/dark cycle (lights on 07:00–19:00). The Ege University Animal Ethics Committee approved all procedures used in this study (2017-102), which was carried out in compliance with Directive 2010/63/EU regarding the use of animals for scientific purposes. Every effort was made to reduce the number of animals used and minimize animal suffering.

### 2.2. Experimental Design

The animals and experimental procedures used in this study were described in more detail in our previous study [36]. The two studies were planned as separate thesis projects and investigated the effects of nicotine consumption and withdrawal on various neuropeptides and their receptors in the mesocorticolimbic system. In accordance with ethical guidelines, the same set of animals was used for both studies to minimize the number of animals used.

Following an acclimation period of one week, 0.2% saccharin (Sigma-Aldrich, St. Louis, MO, USA) in drinking water was given ad libitum to all rats for three days in the first week. Saccharin is used as a bitterness-masking agent in oral nicotine self-administration studies. At the beginning of the second week, rats were assigned at random to the control group (*n* = 10) or the nicotine exposure group (*n* = 30). The nicotine exposure group received forced oral nicotine administration for 12 weeks. Details of the nicotine treatment regimen are provided below. Rats were weighed once a week throughout the study. At the end of week 13, nicotine-exposed rats were allocated randomly to the chronic exposure (no withdrawal; 0-W), 24 h withdrawal (24-W), and 48 h withdrawal (48-W) groups (*n* = 10/group). Forced oral nicotine treatment was continued in the 0-W group until behavioral testing in week 14. The 24-W and 48-W groups were given pure tap water during the 24- and 48 h withdrawal periods, followed by behavioral testing. After behavioral testing, rats were sacrificed by decapitation and trunk blood was collected for the measurement of blood cotinine levels. Figure 1 shows a timeline of the experimental procedures.

Behavioral testing performed in week 14 consisted of the assessment of somatic withdrawal signs, locomotor activity, and immobility. Details of the methods used in behavioral testing and the results are described in our previous study [36]. Consistent with previous reports [37], chronic nicotine administration enhanced the time spent in locomotion, whereas 24 and 48 h withdrawal decreased the time spent in locomotion and increased the time spent immobile. Furthermore, overall somatic withdrawal signs (frequency of gasps and eye blinking, ptosis, shakes, and teeth chattering) were significantly increased in the 48 h withdrawal group compared to the no-withdrawal and control groups.

### 2.3. Forced Oral Nicotine Treatment

Forced oral nicotine treatment was started in the second week of the study (Figure 1). Rats in the nicotine exposure group were provided drinking bottles containing only nicotine dissolved in tap water, while those in the control group were provided drinking bottles containing pure tap water. Rats were given ad libitum access to the fluids in the bottles. Nicotine (Acros Organics, Thermo Fisher Scientific, Waltham, MA, USA, ditartrate dihydrate salt) treatment was initiated at 25 μg/mL (free base) in the second week and increased to 50 μg/mL (free base) in the third week. The rats were maintained at this dose for 11 weeks. The nicotine-containing water and tap water also included 0.2% saccharin in weeks 1, 2, and 3 and 0.1% saccharin in week 4. Saccharin was omitted from the drinking water in week 5 based on previous data showing the rewarding effects of saccharin [38]. The pH of nicotine- and saccharin-containing solutions was 7.4. All fluids in drinking bottles were replenished twice weekly.

A previous study to determine the optimal oral nicotine dose in rats demonstrated that 50 µg/mL nicotine induced blood cotinine concentrations that were comparable to those found in heavy smokers [39]. In our previous study [36], we also showed that the administration of 50 µg/mL nicotine in drinking water for 12 weeks produced a blood cotinine level of 1009.932 ± 66.5 ng/mL (mean ± SEM). This plasma level is at the upper range of heavy smokers [40] and is consistent with the study by Huang et al. [39]. Cotinine levels decreased 24 h after withdrawal (185.1 ± 42.3 ng/mL) and returned to near control levels (32.3 ± 8.0 ng/mL) 48 h after withdrawal. As mentioned above, the present study used the same set of animals included in our previous study by Birdogan et al. [36].

### 2.4. Tissue Dissection

After behavioral testing, the rats were decapitated using a guillotine, the brains were removed, and brain regions of interest were dissected on ice. The dissection technique was described previously by Birdogan et al. [36]. At the beginning of the dissection, four coronal brain slices were prepared. The stereotaxic coordinates indicated in Paxinos and Watson’s rat brain atlas were used to cut the brain slices [41]. The rostral–caudal slice coordinates in relation to the bregma were 5.20–2.20 mm for the first slice, 2.20–0.20 mm for the second slice, −1.40–−3.30 mm for the third slice, and −4.30–−6.30 mm for the fourth slice. An mPFC tissue sample was excised from the first slice and included the infralimbic, prelimbic, and anterior cingulate cortices. Samples of dorsal striatum (DST), NAc, and septum (lateral and medial septal areas) tissue were dissected from the second slice. Medial hypothalamic area (MHA), amygdala (AMG), and hippocampus (HIP) tissue samples were dissected from the third slice. Finally, the VTA was excised from the fourth slice. Dissection of mPFC, septum, MHA, and HIP was performed using curved forceps and scalpel, whereas the dissection of NAc, DST, AMG, and VTA was performed using brain punches. The AMG tissue sample included the CeA and the basomedial and basolateral nuclei. The hypothalamic tissue sample included the arcuate (ARC), ventromedial, dorsomedial (DMN), and paraventricular (PVN) nuclei.

### 2.5. RNA Extraction and cDNA Synthesis

Immediately after dissection, total RNA was extracted from the brain samples and cDNA was generated by reverse transcription. This study used cDNA aliquots that were prepared in our previous study and stored at −20 °C until PCR analysis [36]. The RNA extraction and cDNA synthesis protocols used were described in detail by Birdogan et al. [36]. In brief, Trizol reagent (Invitrogen, Thermo Fisher Scientific, Waltham, MA, USA) was used to extract total RNA from the brain samples. Genomic DNA contamination was removed from the RNA samples by treating with DNase-I (Invitrogen, Thermo Fisher Scientific, Waltham, MA, USA). The RNA level in each sample was quantified by NanoDrop™ ND-2000, a microvolume UV Vis spectrophotometer (Thermo Fisher Scientific, Waltham, MA, USA). To confirm that isolated RNA samples were free of contamination, RNA purity ratios A260/A230 and A260/A280 were calculated. 18S and 28S rRNA bands on a gel containing 1% agarose were stained with ethidium bromide to demonstrate the integrity of the RNA. Samples that showed signs of degradation or contamination were excluded from analysis to ensure data quality. For each sample, total RNA (500 ng) and anchored-oligo(dT) primers were added to the reaction mix, which was prepared as per the manufacturer’s instructions using iScript cDNA Synthesis Kit (Bio-Rad Laboratories, Hercules, CA, USA). The reaction mix was incubated in the thermal cycler at 46 °C for 20 min for the synthesis of first-strand cDNA. Following the denaturation of cDNA chains, tubes were heated at 95 °C for 2 min to inactivate reverse transcriptase. The final concentration of cDNA was adjusted to 5 ng/mL by diluting it 1:5 with nuclease-free water (Bio-Rad Laboratories, Hercules, CA, USA).

### 2.6. Quantitative Real-Time PCR (qPCR)

Expression of CRF, CRFR1, and CRFR2 mRNAs was analyzed in the VTA, MHA, AMG, HIP, septum, DST, NAc, and mPFC. The expression of MC4R mRNA was analyzed in the MHA, AMG, HIP, septum, DST, NAc, and mPFC; and MC3R mRNA in the VTA, MHA, HIP, and septum. Brain areas that contained high MC3R and MC4R mRNA expression were examined by qPCR using the two-step approach. A LightCycler 480 System (Roche Diagnostics, Indianapolis, IN, USA) was used to quantify gene expression levels. The PCR mix was prepared with LightCycler 480 Probes Master (Roche Life Sciences, Indianapolis, IN, USA). Thermal cycling conditions were selected according to the manufacturer’s recommendations for 96-well plates. All samples were run in duplicate to improve the reliability of the results. Beacon Designer 8.14 software was used to design the primers and probes required to amplify the target genes, CRF, CRFR1_,_ CRFR2, MC3R, MC4R, and Actb (Table 1). This study used dual-labeled probes that incorporated Black Hole Quencher 1 and 6-carboxyfluorescein-rhodamine at their 3′ and 5′ ends, respectively. PCR primers specific to the DNA region of interest were designed to span the junctions between exons whenever feasible. All primers were tested by qPCR using five-fold serial dilution of cDNA samples, in order to determine PCR efficiency. The linear correlation coefficient (*R*^2^) determined by the linear regression model was 0.966 for CRF, 0.978 for CRFR1_,_ 0.949 for CRFR2, 0.981 for MC3R, 0.979 for MC4R, and 0.991 for the Actb gene. The following thermocycler conditions were used during data acquisition: 10 min at 95 °C, 40 cycles of 10 s at 95 °C, 30 s at 60 °C, and 1 s at 72 °C. The fold change in the expression level of each gene relative to the internal control (Actb) was evaluated according to the 2^−ΔΔCT^ method.

### 2.7. Statistical Analysis

Group differences in the levels of CRF, CRFR1, CRFR2, MC3R, and MC4R mRNA expression in each brain region were demonstrated by one-way analysis of variation (ANOVA) with Tukey’s post hoc test after verifying that all groups showed normal distribution by D’Agostino–Pearson normality test using GraphPad Prism 5 for Windows (GraphPad Software, Inc., San Diego, CA, USA). Differences in the final body weight measured before sacrifice in the four groups (control, 0-W, 24-W, 48-W) were analyzed with one-way ANOVA followed by Duncan’s post hoc test. The statistical analyses were performed using IBM SPSS Statistics for Windows Version 20.0 (IBM Corp, Armonk, NY, USA), with a *p*-value of ≤0.05 considered statistically significant.

## 3. Results

### 3.1. Effect of Chronic Nicotine Exposure and Withdrawal on Body Weight

In our previous study, statistical analysis of the rats’ body weights measured weekly during chronic oral nicotine treatment showed no significant difference between the control and nicotine-treated groups before the initiation of nicotine administration. However, body weight was lower in the nicotine-treated group than in the controls starting from the first week of nicotine administration and continuing for the duration of treatment. Although the rats’ weight was also monitored during the withdrawal period, the effects of withdrawal on body weight were not analyzed [36]. In the present study, when final body weight measured just before sacrifice at week 14 was analyzed by one-way ANOVA, we observed a significant difference (F_(3,36)_ = 8.737, *p* < 0.001) between the four groups. Duncan’s post hoc test revealed that control animals had significantly higher final weight than rats in the 48-W, 24-W, and 0-W groups, and those in the 48-W group also had a higher final weight compared to the 0-W group (*p* < 0.05; Figure 2).

### 3.2. MC3R and MC4R Transcript Levels following Chronic Nicotine Administration and Withdrawal

The levels of MC3R mRNA expression in the selected brain regions are compared in Figure 3. MC3R mRNA expression in the MHA and VTA showed significant group differences in one-way ANOVA (F_(3,33)_ = 3.592; *p* = 0.024 and F_(3,34)_ = 5.848; *p* = 0.002, respectively). Tukey’s test indicated that 48 h withdrawal increased MC3R mRNA level in the MHA when compared with the control group (*p* = 0.014). The differences between controls and the 0-W or 24-W groups were not significant. MC3R mRNA levels in the VTA were upregulated in the 24-W (*p* = 0.01) and 48-W (*p* = 0.003) groups compared to controls but did not differ significantly between controls and the 0-W group. Chronic nicotine exposure or withdrawal had no effect on MC3R mRNA expression in the septum. Because of the low and unreliable MC3R mRNA levels determined by qPCR analysis of the HIP, statistical analysis of MC3R levels in this brain region was not performed.

The MC4R mRNA expression levels of the groups are compared in Figure 4. One-way ANOVA demonstrated significant differences in the expression of MC4R mRNA in the mPFC (F_(3,33)_ = 5.156; *p* = 0.005), NAc (F_(3,34)_ = 3.239; *p* = 0.034), DST (F_(3,34)_ = 3.123; *p* = 0.039), and septum (F_(3,32)_ = 3.254; *p* = 0.034). Post hoc comparisons using Tukey’s test demonstrated that MC4R mRNA expression in the mPFC was upregulated in the 24-W (*p* = 0.038) and 48-W (*p* = 0.007) groups compared to controls. In contrast, MC4R mRNA levels in the NAc were upregulated in the 0-W group when compared with controls (*p* = 0.042). In the DST, MC4R mRNA expression was higher in the 24-W group than in the control group (*p* = 0.022), whereas septum expression levels were higher in the 48-W group compared to controls (*p* = 0.03). No differences in MC4R mRNA expression were observed in the HIP, AMG, or MHA.

### 3.3. CRF and CRFR Transcript Levels following Chronic Nicotine Administration and Withdrawal

CRF mRNA levels in the selected brain regions are compared in Figure 5. One-way ANOVA revealed significant differences between the groups in the mPFC (F_(3,34)_ = 15.171; *p* < 0.001) and AMG (F_(3,34)_ = 4.474; *p* = 0.009). CRF mRNA levels in the mPFC were elevated in the 0-W, 24-W, and 48-W groups when compared with controls (*p* < 0.001 for all), while levels in the AMG were higher only in the 0-W group than in the control group (*p* = 0.005). Although the results of one-way ANOVA also yielded significant results in the septum (F_(3,33)_ = 2.976; *p* = 0.046) and MHA (F_(3,33)_ = 2.952; *p* = 0.047), Tukey’s test revealed no significant differences in CRF mRNA expression between the study groups. No changes in CRF mRNA expression were detected in the NAc, DST, or VTA after chronic oral nicotine administration or withdrawal.

The CRFR1 mRNA levels are compared in Figure 6. One-way ANOVA showed significant differences between groups in the AMG (F_(3,34)_ = 4.022, *p* = 0.015) and the septum (F_(3,34)_ = 3.006, *p* = 0.044). Tukey’s post hoc test indicated that CRFR1 mRNA expression in the AMG was significantly elevated in the 0-W (*p* = 0.02) and 48-W (*p* = 0.028) groups when compared to controls, but the increase in the 24-W group did not reach significance. In the septum, CRFR1 mRNA expression was upregulated in the 48-W group compared to the control group (*p* = 0.045). In the NAc, one-way ANOVA suggested a significant difference in CRFR1 mRNA (F_(3,34)_ = 2.995, *p* = 0.044), but the results of post hoc analyses were not significant. CRFR1 mRNA expression in the VTA, MHA, DST, and mPFC was not significantly affected by chronic nicotine exposure and withdrawal.

In the analysis of CRFR2 mRNA expression, there were no significant differences between groups in any of the brain regions studied. The CRF, CRFR1, and CRFR2 mRNA levels determined by qPCR were very low and unreliable in the HIP. Therefore, these data were not statistically analyzed.

## 4. Discussion

In our previous study, we showed that nicotine-withdrawn rats displayed an increase in total somatic signs at 48 h [36]. Also, rats withdrawn for 24 and 48 h spent less time in locomotion and more time immobile. Studies suggest that such behavioral changes observed during nicotine withdrawal resemble a depression-like state [2]. In this study, we used cDNA samples prepared from these nicotine-treated rats and demonstrated that chronic nicotine consumption and withdrawal regulated the expression of CRF, CRFR1, MC3R, and MC4R mRNA in the mesocorticolimbic system.

In this study, MC3R mRNA expression was found to be increased after 24 and 48 h withdrawal, which suggests that MC3R signaling in the VTA may regulate withdrawal-induced negative effects. Despite the presence of numerous studies showing the effect of MC3Rs on reward, data concerning the effect of MC3Rs on negative effects are limited. One study by Sweeney et al. [42] demonstrated that MC3R knockout male mice displayed increased anxiety-like behavior in the EPM and open field tests. However, the site-specific effect of MC3R signaling in the VTA on negative effects is not known. Both DA and non-DA neurons in the VTA express MC3R [43]. Injections of α-MSH into the VTA have been consistently found to increase DA levels in the NAc [17] and enhance behaviors such as grooming and locomotion, which depend on DA [18]. Additionally, α-MSH increased the firing rate of MC3R-expressing DA and non-DA neurons in the VTA [44]. In parallel, we previously demonstrated that nicotine injections for 6 days elevated MC3R mRNA expression in the VTA, supporting the idea that MC3R signaling may contribute to nicotine reward during a short-term nicotine exposure regimen [45]. Interestingly, the present study showed that chronic oral nicotine exposure for 12 weeks did not affect MC3R mRNA expression in the VTA, which suggests that MC3R signaling does not contribute to reward during chronic nicotine exposure. Previously, Dunigan et al. [46] demonstrated that the NAc, bed nucleus of the stria terminalis, lateral septum, lateral habenula, and basolateral nucleus are bidirectionally connected to MC3R-expressing neurons in the VTA. These brain regions have been implicated in the regulation of anxiety and aversive behaviors, as well as reward [47,48,49,50]. Furthermore, non-DA neurons in the VTA that project to the lateral habenula or NAc are suggested to contribute to aversive behavior [51]. Studies also indicate another aversion-inducing pathway that involves a distinct set of VTA DA neurons receiving glutamatergic inputs from lateral habenula and projecting to the mPFC [48]. MC3Rs expressed on such non-DA and DA neurons in VTA may contribute to the aversive state triggered during nicotine withdrawal. This is supported by studies showing that intra-VTA injection of a melanocortin receptor agonist reduced 24 h food intake, consumption of sucrose solution in the two-bottle choice test, and operant responding for sucrose pellets [52,53,54].

MC3R mRNA is expressed at a high level in the ARC and ventromedial nuclei of the MHA [11]. In the ARC, MC3R is located on POMC- and Agouti-related peptide/neuropeptide Y neurons, where it may function as an autoreceptor exerting an inhibitory effect on POMC signaling [55,56,57]. Previous studies demonstrated that ICV administration of MC3R agonists and activation of MC3R-expressing ARC neurons increased food intake and body weight [42,58]. Nicotine withdrawal is also known to stimulate food consumption and cause weight gain [59]. In this study, 48 h withdrawal increased both body weight and MC3R mRNA levels in the MHA. It is possible that upregulation of MC3R mRNA levels in the MHA underlies the weight gain observed during nicotine withdrawal. However, neither MC3R nor MC4R mRNA expression in the hypothalamus changed during chronic nicotine exposure. A previous study also showed that nicotine injections for 2 or 9 days did not alter melanocortin receptor mRNA levels in the PVN, DMN, or ARC [60].

In this study, chronic oral nicotine treatment also upregulated MC4R mRNA in the NAc, whereas 24 and 48 h withdrawal increased MC4R mRNA in the septum, DST, and mPFC. In the striatum, MC4R co-expression is found primarily in prodynorphin and D1 receptors expressing medium spiny neurons [61,62]. Currently, the identity of the neuron populations that express MC4R in the mPFC and septum is not clear. Some studies suggest that MC4Rs may mediate the rewarding effects of drugs. A previous study [61] reported that the injection of an MC4R antagonist into the NAc blocked cocaine self-administration and cocaine-induced locomotor response and place conditioning. In parallel, locomotor sensitization to repeated cocaine exposure was decreased in MC4R-null mice, and re-expressing MC4Rs in D1 receptors containing neurons restored the locomotor sensitization [63]. On the other hand, numerous studies have also demonstrated that signaling through MC4Rs causes depression-like, anxiogenic, and anti-reward effects [64], while ICV injection of MC4R antagonists reduces depressive and anxiety-like behaviors [13,65]. Kokare et al. [14] also showed that ethanol withdrawal-induced anxiety was mitigated by ICV injection of a selective MC4R antagonist. However, the behavioral consequences of MC4R-mediated signaling in the reward system during nicotine withdrawal are not clear. Only one study by Qi et al. [16] showed that ICV administration of selective MC4R antagonists prevented the reinstatement of nicotine seeking in response to stress. Based on our current results, we suggest that increased MC4R signaling in the mPFC, DST, and septum during nicotine withdrawal may be a contributing factor in the negative affective state triggered by withdrawal. Considering the possible role of accumbal MC4Rs in cocaine reward, we can also propose that increased MC4R signaling in the NAc during nicotine exposure may regulate nicotine reward. Previously, we showed that nicotine injections for 6 days upregulated MC4R mRNA levels in the mPFC during nicotine reward [45]. However, the present study indicates that consumption of nicotine in drinking water for 12 weeks had no effect on MC4R mRNA expression in the mPFC, whereas an increase was detected during nicotine withdrawal. MC4R activity in different neuron populations in the mPFC may regulate reward during acute nicotine exposure and negative effects during withdrawal.

In the present study, chronic nicotine exposure caused increased CRF and CRFR1 mRNA expression in the AMG and increased CRF mRNA expression in the mPFC. The increases in CRFR1 mRNA in the AMG and CRF mRNA in the mPFC also persisted during withdrawal. In addition, CRFR1 mRNA in the septum was increased only by nicotine withdrawal. During nicotine exposure and withdrawal, the expression and function of nicotinic, glutamatergic, and dopaminergic receptors show significant differential changes in subregions of the reward system [1]. Therefore, changes in the expression or function of these receptors may regulate neuropeptide expression differentially in different brain regions. Overall, increased CRF signaling, possibly through CRFR1 in the AMG, septum, and mPFC, may have an important role in promoting the negative affective state associated with nicotine withdrawal. CRF signaling via CRFR1 in the AMG, septum, and PFC has been implicated in negative affective states such as anxiety and aversion. CRF neurons in the basolateral nucleus and mPFC project axons to the NAc [66], whereas CRF neurons in the PVN and CeA send projections to the VTA [67]. CRFR1 is localized on DA neurons in the VTA [68] and DA terminals in the NAc [69]. CRF signaling through CRFR1 has a regulatory effect on VTA DA neuron activity and on DA release in the NAc and PFC, although some studies reported that this effect caused a reduction [70], while others indicated an increase [71]. Changes in mesocorticolimbic DA system activity are associated with reward, aversion, and anxiety-like behavior [21,68,72]. Many studies have shown that CRF signaling in the AMG [73], NAc [74], septum [25], and PFC [27,28] is related to anxiogenic-like and avoidance behaviors. In parallel, administering nonspecific CRFR antagonists into the NAc shell and CeA or selective CRFR1 antagonists into the CeA prevented elevations in brain reward thresholds induced by mecamylamine in the nicotine-dependent rats [75,76]. Interestingly, in another study by Qi et al. [77], CRF overexpression accompanied by a decrease in CRFR1 mRNA levels in the CeA diminished withdrawal-induced elevations in the intracranial self-stimulation test threshold in rats treated with nicotine.

In contrast, there are some studies indicating the presence of CRF neuron populations in the CeA and dorsomedial PFC that suppress depressive- and anxiety-like behaviors [21,78]. Interestingly, Lemos et al. [69] showed that CRF administration in the NAc induced preference for the CRF-paired chamber. In another study [79], optogenetic stimulation of CRF neurons in the CeA amplified incentive motivation (wanting) for sucrose and increased activity in the mesocorticolimbic reward system. CRF signaling in different neuron populations can exert opposing effects on affect under different conditions. Therefore, it is possible that CRF signaling in specific neural pathways induces reward or positive effects during nicotine exposure, while CRF signaling in other neural pathways triggers negative effects during stress, such as withdrawal.

Chronic nicotine exposure or withdrawal has been reported to increase CRF and CRFR1 mRNA expression in the VTA [30], DST [34], and AMG [29,35]. In our study, there was no change in the VTA or DST. Differences in nicotine regimen and species may explain the discrepancies in the results of different studies. On the other hand, some studies demonstrated no change in CRF and CRFR1 mRNA levels in the NAc of adult male rats, which is consistent with the findings in this study [29,35]. Additionally, our study showed that neither nicotine exposure nor withdrawal changed CRF mRNA expression in the hypothalamus. In parallel, Semba et al. [80] showed that expression of CRF mRNA in the PVN was comparable in the withdrawal group and controls. To the best of our knowledge, this study is the first to show that CRF and CRFR1 mRNA expressions are regulated in the mPFC and septum during nicotine withdrawal.

## 5. Study Limitations

This study measured only the mRNA expression of CRF, CRF1, MC3R, and MC4R in specific brain regions. However, changes in mRNA levels do not always correlate to changes in the protein levels in tissues. Therefore, future studies should be conducted to test if the alterations in mRNA levels were reflected at the protein level. Furthermore, in this study, we examined the effects of acute nicotine withdrawal (24–48 h) on gene expression. Further studies can investigate the effects of protracted abstinence.

Finally, although all available measures were taken to prevent degradation during RNA isolation and avoid contamination of the RNA with DNA and protein, some samples showed signs of degradation or contamination and were excluded from analysis. Additionally, when there were major differences between replicates of qPCR results, those samples were also excluded. These measures reduced the number of samples included in some comparisons. Future studies including a higher number of animals can be conducted to further elucidate the significant changes observed.

## 6. Conclusions

In summary, the present study showed that the expression of MC3R, MC4R, CRF, and CRFR1 mRNAs in the mesocorticolimbic system is increased by chronic nicotine exposure and withdrawal. Based on these findings, we suggest that increased melanocortin signaling through MC3Rs and MC4Rs and increased CRF signaling through CRFR1 may regulate reward and the symptoms of nicotine withdrawal. This is the first study which showed that melanocortin receptor mRNA expression in the mesocorticolimbic system is regulated by chronic nicotine consumption and withdrawal. However, further studies are required to test whether these alterations in mRNA levels are translated into alterations in protein levels. Future investigations directed toward understanding the effects of melanocortin and CRFR1 signaling on behavior and effects during nicotine dependence will provide valuable information for developing new targets in smoking cessation treatment.

## Figures and Tables

**Figure 1 brainsci-14-00063-f001:**
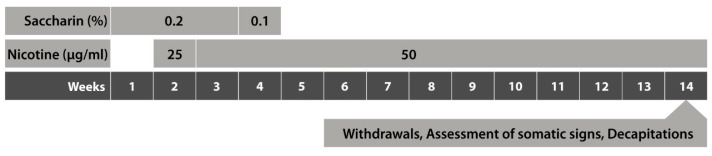
Timeline of the experimental design.

**Figure 2 brainsci-14-00063-f002:**
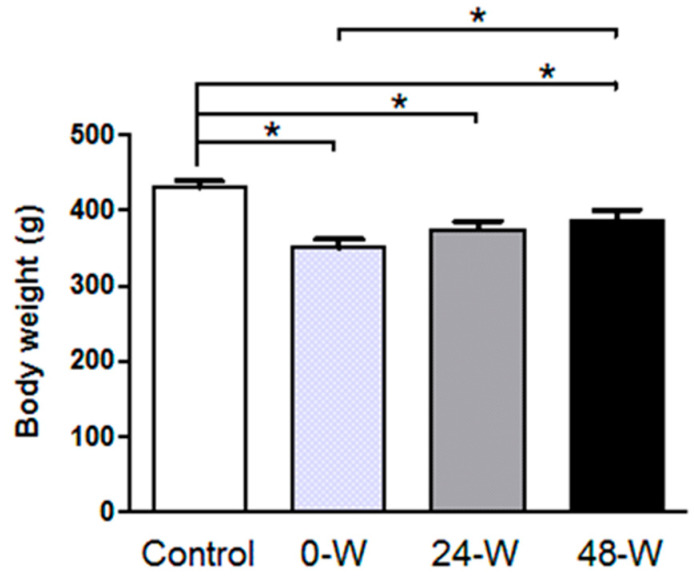
Effect of nicotine consumption in water for 12 weeks and withdrawal on the body weight of rats measured before sacrifice. Body weight was lower after chronic nicotine exposure (no withdrawal, 0-W), 24 h withdrawal (24-W), and 48 h withdrawal (48-W) compared to controls. Bars represent the mean ± standard error of mean; * *p* < 0.05 (*n* = 10 per group).

**Figure 3 brainsci-14-00063-f003:**
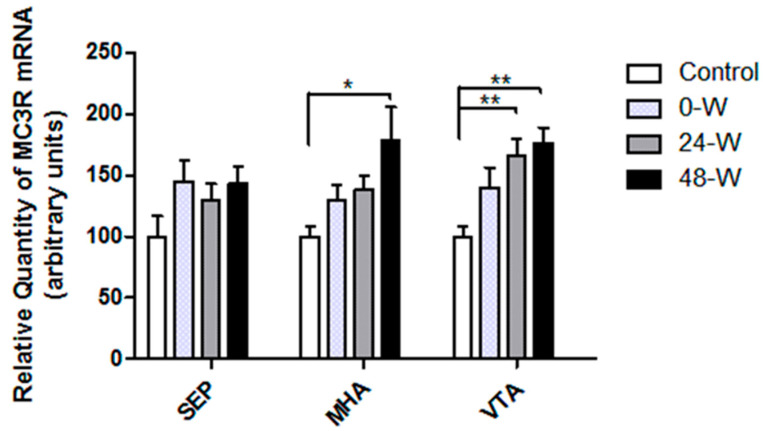
Comparison of MC3R mRNA levels in septum (SEP), medial hypothalamic area (MHA) and the ventral tegmental area (VTA), and in rats following chronic nicotine exposure (0-W), 24 h withdrawal (24-W), and 48 h withdrawal (48-W). Bars represent the mean ± standard error of mean; * *p* < 0.05, ** *p* ≤ 0.01 (*n* = 8–10 per group).

**Figure 4 brainsci-14-00063-f004:**
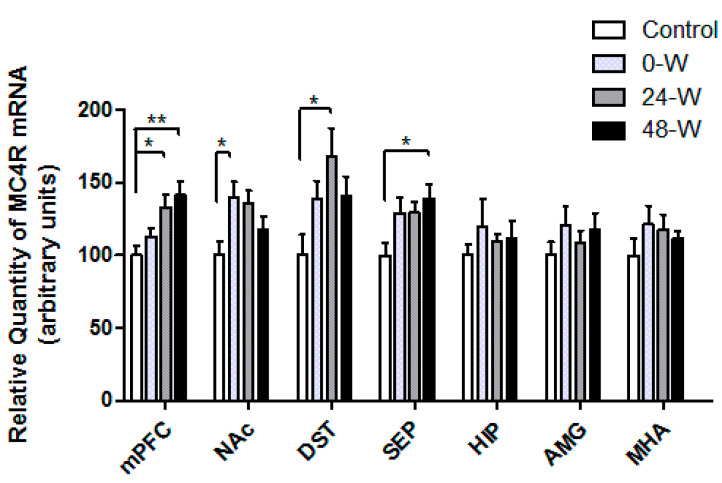
MC4R mRNA levels in the medial hypothalamic area (MHA), amygdala (AMG), hippocampus (HIP), septum (SEP), dorsal striatum (DST), nucleus accumbens (NAc), and medial prefrontal cortex (mPFC) in rats following chronic nicotine exposure (0-W), 24 h withdrawal (24-W), or 48 h withdrawal (48-W). Bars represent the mean ± standard error of mean; * *p* < 0.05, ** *p* < 0.01 (*n* = 8–10 per group).

**Figure 5 brainsci-14-00063-f005:**
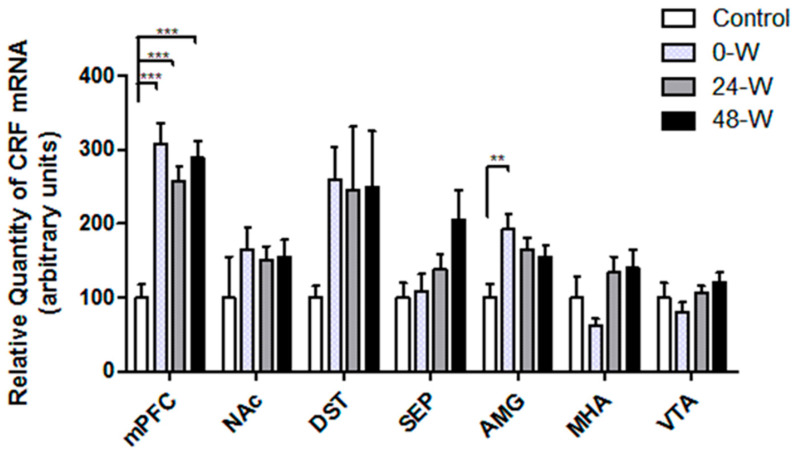
Comparison of CRF mRNA levels in the ventral tegmental area (VTA), medial hypothalamic area (MHA), amygdala (AMG), septum (SEP), dorsal striatum (DST), nucleus accumbens (NAc), and medial prefrontal cortex (mPFC) in rats following chronic nicotine exposure (0-W), 24 h withdrawal (24-W), and 48 h withdrawal (48-W). Bars represent the mean ± standard error of mean; ** *p* < 0.01, *** *p* < 0.001 (*n* = 7–10 per group).

**Figure 6 brainsci-14-00063-f006:**
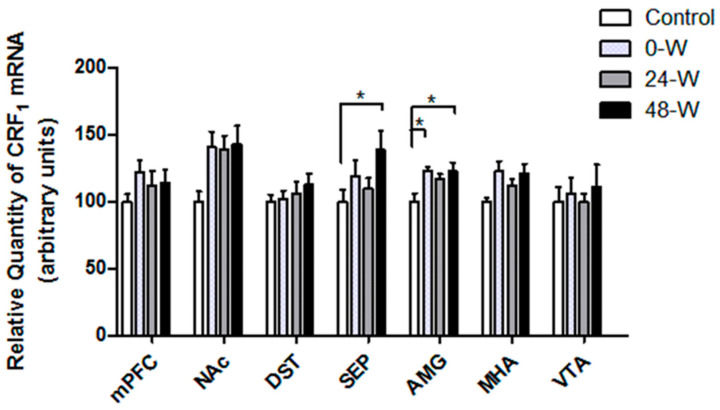
Comparison of CRFR1 mRNA levels in the ventral tegmental area (VTA), medial hypothalamic area (MHA), amygdala (AMG), septum (SEP), dorsal striatum (DST), nucleus accumbens (NAc), and medial prefrontal cortex (mPFC) in rats following chronic nicotine exposure (0-W), 24 h withdrawal (24-W), or 48 h withdrawal (48-W). Bars represent the mean ± standard error of mean; * *p* < 0.05 (*n* = 8–10 per group).

**Table 1 brainsci-14-00063-t001:** Primer and probe sequences used in the qPCR.

Gene	Sense	Antisense	Probe
CRF (NM_031019)	cccaagtacgttgagaaa	ctctcttctcctcccttg	tgagcccgcactgttgttct
CRF_1_ (NM_001301812)	gtccagtcaggagataac	gcagaatagtaagagtctaatg	agcattatcagaccgcactcca
CRF_2_ (NM_022714)	gaaggtccctactcctac	atgccgttgaagtattcg	caacacgaccttggaccagat
MC3R (NM_001025270)	gccgataaccatgaactc	tctggcttgatgaaaacc	cctcttatccgacgctgcctaa
MC4R (NM_013099)	gagtgaatactacggctaa	tgctcatcatttctttagaag	ctctcggctgaccagtctgc
Actb (NM_031144)	cccgcgagtacaaccttct	cgtcatccatggcgaact	agctcctccgtcgccggtcca

## Data Availability

The data presented in this study are available on reasonable request from the corresponding author. The data are not publicly available because dissemination has yet to be explicitly approved by the local ethics committee.

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
