# Peer review of "Chronic Nicotine Consumption and Withdrawal Regulate Melanocortin Receptor, CRF, and CRF Receptor mRNA Levels in the Rat Brain"

_brainsci, 2024, doi:10.3390/brainsci14010063_

Round 1
Reviewer 1 Report
Comments and Suggestions for Authors
Nicotine, by affecting nicotinic cholinergic receptors in the brain that ale involved in release of dopamine and other neurotransmitters, is responsible for smoking addiction, which is the main cause of disability and premature death. The secretion of dopamine, glutamate and GABA is particularly important in the development of nicotine addiction, and CRF may play a key role in the withdrawal syndrome. A better understanding of the mechanisms of nicotine addiction leads to the development of new drugs (e.g. varenicline) that act on specific subtypes of nicotinic receptors. The development of other drugs that act on nicotinic receptors and other mediators associated with smoking addiction may lead to further improvements in the effectiveness of pharmacotherapy leading to smoking cessation. Therefore, I consider the topic discussed by the Authors to be extremely important and still actual. Additionally, not all aspects of nicotine addiction have been fully explained therefore the Autorzy examined the effects of chronic oral nicotine exposure and withdrawal on the mRNA expressions of melanocortin receptors , CRF 83 and CRF receptors in the mesocorticolimbic system which can significantly expand existing knowledge.
The manuscript was written in accordance with the requirements for this type of work and the requirements of the journal. The abstract is a brief summary of all elements of the manuscript. The introduction presents all necessary information and ends with a well-formulated goal of the experiments. The methodology and results are written in a precise, understandable way. The discussion explains all the research conducted by the Authors and formulates basic explanations of the observed effects. The summary also accurately defines the Authors' future research goals.
I only suggest that the Authors should add a limitations section and ask them to explain why rats of such an age were used and why there are different numbers of rats in the ANOVA values?
Author Response
Reviewer 1:
We would like to thank the reviewer for their positive and constructive feedback. Below are our point-by-point responses.
Comment 1: I only suggest that the Authors should add a limitations section.
Response 1: We added a study limitations section before the conclusion section of the manuscript.
Comment 2: I ask them to explain why rats of such age were used.
Response 2: We aimed to study the effects of nicotine in adult animals. We clarified this point by revising the “Animals” section. Rats are considered adults starting at 3 months of age. Their weight also reaches 300 g by this time, which is an important threshold where rapid weight gain and growth plateau.
Comment 3: Why there are different numbers of rats in the ANOVA values?
Response 3: We thank the reviewer for their careful attention. We added an explanation as to why different numbers of samples were included in the ANOVA.
Although all groups included 10 rats each and no animals were lost, we excluded a few samples from the analysis to ensure data quality. As mentioned in the methods section, RNA samples were run on a spectrophotometer to evaluate the quantity and the purity of extracted RNA using optical density ratios of 260/280 and 260/230. The RNA integrity was also confirmed by ethidium bromide staining of 28S and 18S ribosomal RNA bands on 1% agarose gel. RNA samples showing degradation were excluded from further experiments. We added a statement explaining this in the “RNA Extraction and cDNA Synthesis” section. Furthermore, when there were major differences between replicates of qPCR results, those samples were also excluded. Due to the large numbers of regions, samples, and targets in our study, we chose not to specify numbers for each group because it may be confusing for readers.
Reviewer 2 Report
Comments and Suggestions for Authors
In this study, authors evaluated the effects of chronic oral nicotine exposure and withdrawal on the mRNA expressions of MC3R and MC4R, CRF, and CRF receptors in the mesocorticolimbic system. The authors showed that after withdrawal rats increased body weight almost as the same as controls, and they also found that expression of CRF and CRFR1 increased in both with and without withdrawal. An important finding was that just in the withdrawal group was increased the expression of melanocortin receptors, while in the chronic group increased MC4R expression. Together these findings suggest that nicotine exposure with or without withdrawal modulate CRF and melanocortin receptors that could be associated to negative affective state. This study is interesting; however, the authors should be considered next points before this manuscript can be taken for publication.
1. How do the authors explain that 48 hours withdrawal rats increase body weight? This effect is so fast.
2. In the statistical analysis, why do the authors consider that they groups have a normal distribution? Why don’t they use a nonparametric test?
3. Why the number of samples are different between tests, for body weight 10 animals, and for PCR are used 7-10 or 8-10?
4. Authors should add a figure that resumes their findings.
Author Response
Reviewer 2:
We thank the reviewer for their time and constructive feedback. Below are our point-by-point responses to their questions.
Comment 1: How do the authors explain that 48 hours withdrawal rats increase body weight? This effect is so fast.
Response 1: We agree with the reviewer that the effect of nicotine withdrawal on body weight is rapid in rats. Our results are in accordance with previous studies reporting that nicotine withdrawal for 24, 48, and 72 hours resulted in weight gain (Malin et al. 1992; Biala and Weglinska 2005; Wellman et al. 2005; Fornari et al. 2007). This rapid weight gain during nicotine withdrawal may be explained by increased food consumption during acute nicotine withdrawal.
Malin, D.H., & Goyarzu, P. Rodent models of nicotine withdrawal syndrome. Handbook of experimental pharmacology 2009, 401-434.
Biala, G., & Weglinska, B. (2005). Blockade of the expression of mecamylamine-precipitated nicotine withdrawal by calcium channel antagonists. Pharmacological research, 51(5), 483–488.
Wellman, P.J., Bellinger, L.L., Cepeda-Benito, A., Susabda, A., Ho, D.H., & Davis, K.W. (2005). Meal patterns and body weight after nicotine in male rats as a function of chow or high-fat diet. Pharmacology, biochemistry, and behavior, 82(4), 627–634.
Fornari, A., Pedrazzi, P., Lippi, G., Picciotto, M. R., Zoli, M., & Zini, I. (2007). Nicotine withdrawal increases body weight, neuropeptide Y and Agouti-related protein expression in the hypothalamus and decreases uncoupling protein-3 expression in the brown adipose tissue in high-fat fed mice. Neuroscience letters, 411(1), 72–76.
Comment 2: In the statistical analysis, why do the authors consider that they groups have a normal distribution? Why don't they use a nonparametric test?
Response 2: We thank the reviewer for raising this point. This was an oversight on our part while reporting our statistical methods. Before conducting one-way ANOVA, we checked our data for normal distribution using the D’Agostino-Pearson normality test. We revised the “Statistical Analysis” section accordingly and described the methods used.
Despite the low sample size, there is evidence for the robustness of the ANOVA even in the presence of heteroscedasticity and non-normal distributions (Blanca et al. 2017; Levine & Hullet 2002). Although it is controversial whether or not a parametric test such as one-way ANOVA with F test can be utilized with non-normally distributed data, we still believe normality should be checked if groups include fewer than 25 samples.
Blanca M.J., Alarcón R., Arnau J., Bono R., Bendayan R. Non-Normal Data: Is ANOVA Still a Valid Option? Psicothema. 2017;29:552–557.
Levine T.R., Hullet C.R. Eta Squared, Partial Eta Squared, and Misreporting of Effect Size in Communication Research. Hum. Commun. Res. 2002;28:612–625.
Comment 3: Why the number of samples are different between tests, for body weight 10 animals, and for PCR are used 7-10 or 8-10?
Response 3: We thank the reviewer for their careful attention. We added an explanation as to why different numbers of samples were included in the ANOVA. All groups included 10 rats and no animals were lost during the study period. The animals and experimental procedures used in this study were described in more detail in our previous study (Birdogan et al. 2021). However, we excluded a few samples from the analysis to ensure data quality. As mentioned in the methods section, RNA samples were run on a spectrophotometer to evaluate the quantity and the purity of extracted RNA using optical density ratios of 260/280 and 260/230. The RNA integrity was also confirmed by ethidium bromide staining of 28S and 18S ribosomal RNA bands on 1% agarose gel. RNA samples showing degradation were excluded from further experiments. We added a statement explaining this in the “RNA Extraction and cDNA Synthesis” section. Furthermore, when there were major differences between replicates of qPCR results, those samples were also excluded. Due to the large numbers of regions, samples, and targets in our study, we chose not to specify numbers for each group because it may be confusing for readers.
Birdogan, A.; Salur, E.; Tuzcu, F.; Gokmen, R.C.; Ozturk Bintepe, M.; Aypar, B.; Keser, A.; Balkan, B.; Koylu, E.O.; Kanit, L.; et al. Chronic oral nicotine administration and withdrawal regulate the expression of neuropeptide Y and its receptors in the mesocorticolimbic system. Neuropeptides 2021, 90, 102184.
Comment 4: Authors should add a figure that resumes their findings.
Response 4: We thank the reviewer for encouraging us to prepare a graphical abstract which will better engage the reader with our manuscript. We are submitting a graphical abstract that summarizes our key findings.
Reviewer 3 Report
Comments and Suggestions for Authors
In the manuscript titled ‘Chronic Nicotine Consumption and Withdrawal Regulate Melanocortin Receptor, CRF, and CRF Receptor mRNA Levels in the Rat Brain’, the authors found that chronic nicotine consumption and withdrawal regulated the mRNA levels of MCR, CRF and CRF receptors. The results are interesting. I only have few comments here.
1, 24h and 48 h withdrawal caused different increasing of MC3R, MC4R, CRF and CRFR. What would this tell us? The nicotine withdrawal symptoms in rodents are most pronounced within the first week after cessation of nicotine exposure, with mouse peaked in around 5 days and rats peaked in around 3 days. The, what would be the reason the authors choose to test 24h and 48 hours? What would be the results if the authors tested the mRNA levels in 72 hours or after one week of withdrawal period?
2, The nicotine must enter the CNS to have its effects. The authors should report the serum and brain concentration of nicotine.
3, The body weight of rodents is changing along time. So, it would be great if the author reports the bodyweight comparison each week.
4, CRF mRNA was increased in mPFC and AMG, CRFR mRNA was increased in SEP and AMG. What would be the underlying mechanisms that caused the CRF and its receptors changed in different brain regions?
Author Response
We would like to thank the reviewer for their positive and constructive feedback. Below are our point-by-point responses.
Comment 1: 24h and 48h withdrawal caused different increasing of MC3R, MC4R, CRF and CRFR. What would this tell us? The nicotine withdrawal symptoms in rodents are most pronounced within the first week after cessation of nicotine exposure, with mouse peaked around 5 days and rats peaked around 3 days. The, what would be the reason the authors choose to test 24h and 48 hours? What would be the results if the authors tested the mRNA levels in 72 hours or after one week of withdrawal period?
Response 1: In this study we examined the acute effects of nicotine withdrawal (24-48 hours) on gene expression. However, the effects of protracted abstinence can be studied in future experiments.
Differences in the upregulation of CRF, CRF, and melanocortin receptor mRNAs induced by 24- and 48-hour withdrawal may reflect differences in the molecular mechanisms causing increased expression of these genes.
In our previous paper by Birdogan et al. (2021) reporting the somatic withdrawal signs and locomotor activity obtained from the same animals, we observed that the overall somatic withdrawal signs were increased in the 48-W group compared to the 0-W and control groups and the duration of locomotor activity decreased and the duration of immobility increased in the 24-W and 48-W groups compared to the 0-W group. Therefore, we chose to analyze the effect of acute withdrawal on mRNA expression levels at these time points.
Birdogan, A.; Salur, E.; Tuzcu, F.; Gokmen, R.C.; Ozturk Bintepe, M.; Aypar, B.; Keser, A.; Balkan, B.; Koylu, E.O.; Kanit, L.; et al. Chronic oral nicotine administration and withdrawal regulate the expression of neuropeptide Y and its receptors in the mesocorticolimbic system. Neuropeptides 2021, 90, 102184.
Comment 2: The nicotine must enter the CNS to have its effects. The authors should report the serum and brain concentration of nicotine.
Response 2: We agree with the reviewer on this. We believe it is always necessary to check the amount of nicotine received by the animal whenever a self-administration treatment model is used. However, the in vivo and in vitro measurement of nicotine levels requires high-pressure liquid chromatography (HPLC) and the half-life of nicotine is very short (minutes). Therefore, it is common practice to measure levels of cotinine, which is the main metabolite of nicotine and stable for much longer periods of time. In this study, we used a self-administration method (forced oral nicotine) for treatment ad libitum and measured serum cotinine levels using ELISA. As summarized in the last paragraph of the “Forced Oral Nicotine Treatment” section, oral nicotine treatment yielded plasma cotinine levels of 1009.932 ± 66.5 ng/ml (mean±SEM). This plasma level is at the upper range of heavy smokers. Cotinine levels decreased 24 hours after withdrawal (185.1 ± 42.3 ng/ml) and returned to near control levels (32.3 ± 8.0 ng/ml) at 48 hours after withdrawal. These results indicate that nicotine is present in the blood and therefore is distributed throughout the body via the circulation. Nicotine readily crosses the blood-brain barrier and binds with high efficiency to brain tissue as reviewed by Matta et al. (2007).
The animals and experimental procedures used in this study were described in more detail in our previous study (Birdogan et al. 2021). The two studies were planned as separate thesis projects and investigated the effects of nicotine consumption and withdrawal on various neuropeptides and their receptors in the mesocorticolimbic system. In accordance with ethical guidelines, the same set of animals was used for both studies to minimize the number of animals used. As the changes in blood cotinine levels during chronic oral nicotine administration and withdrawal of the animals used in our study were already published in Birdogan et al. (2021), we only summarized them in the current manuscript to avoid reporting previously published data.
Matta, S. G., Balfour, D. J., Benowitz, N. L., Boyd, R. T., Buccafusco, J. J., Caggiula, A. R., Craig, C. R., Collins, A. C., Damaj, M. I., Donny, E. C., Gardiner, P. S., Grady, S. R., Heberlein, U., Leonard, S. S., Levin, E. D., Lukas, R. J., Markou, A., Marks, M. J., McCallum, S. E., Parameswaran, N., … Zirger, J. M. (2007). Guidelines on nicotine dose selection for in vivo research. Psychopharmacology, 190(3), 269–319.
Birdogan, A.; Salur, E.; Tuzcu, F.; Gokmen, R.C.; Ozturk Bintepe, M.; Aypar, B.; Keser, A.; Balkan, B.; Koylu, E.O.; Kanit, L.; et al. Chronic oral nicotine administration and withdrawal regulate the expression of neuropeptide Y and its receptors in the mesocorticolimbic system. Neuropeptides 2021, 90, 102184.
Comment 3: The body weight of rodents is changing along time. So, it would be great if the author reports the body weight comparison each week.
Response 3: We revised the “Effect of chronic nicotine exposure and withdrawal on body weight” section to clarify this point. As the reviewer suggested, we already reported weekly weight changes observed during nicotine treatment (1-12 weeks) in our previous paper by Birdogan et al. (2021). On the other hand, in this study we are reporting the effect of withdrawal on body weight, which we have not previously analyzed or reported.
Birdogan, A.; Salur, E.; Tuzcu, F.; Gokmen, R.C.; Ozturk Bintepe, M.; Aypar, B.; Keser, A.; Balkan, B.; Koylu, E.O.; Kanit, L.; et al. Chronic oral nicotine administration and withdrawal regulate the expression of neuropeptide Y and its receptors in the mesocorticolimbic system. Neuropeptides 2021, 90, 102184.
Comment 4: CRF mRNA was increased in mPFC and AMG, CRFR mRNA was increased in SEP and AMG. What would be the underlying mechanisms that caused the CRF and its receptors changed in different brain regions?
Response 4: We revised the discussion section to address the possible underlying mechanisms of differential gene expression. During nicotine exposure and withdrawal, the expression and function of nicotinic, glutamatergic, and dopaminergic receptors show significant differential changes in subregions of the reward system (Pistillo et al. 2015). Therefore, changes in the expression or function of these receptors may regulate neuropeptide expression differentially in different brain regions.
Pistillo, F.; Clementi, F.; Zoli, M.; Gotti, C. Nicotinic, glutamatergic and dopaminergic synaptic transmission and plasticity in the mesocorticolimbic system: focus on nicotine effects. Progress in neurobiology 2015, 124, 1-27.
Round 2
Reviewer 2 Report
Comments and Suggestions for Authors
Authors considered the reviewer's comments and the manuscript improve.
Reviewer 3 Report
Comments and Suggestions for Authors
The authors solved all my concerns.